# Effects of Geographic Region on the Composition of Bactrian Camel Milk in Mongolia

**DOI:** 10.3390/ani9110890

**Published:** 2019-11-01

**Authors:** Jing He, Yuchen Xiao, Khongorzul Orgoldol, Liang Ming, Li Yi, Rimutu Ji

**Affiliations:** 1Key Laboratory of Dairy Biotechnology and Bioengineering, Ministry of Education, College of Food Science and Engineering, Inner Mongolia Agricultural University, Hohhot 010018, China; hejing1409@163.com (J.H.); xycimau@163.com (Y.X.); khongoroo19@163.com (K.O.); bmlimau@163.com (L.M.); yili_imau@163.com (L.Y.); 2Camel Research Institute of Inner Mongolia, Alxa 737300, China

**Keywords:** Bactrian camel, differentiation, camel milk, gross composition, fatty acids, amino acids

## Abstract

**Summary:**

Camels are known to occupy arid and desert countries. These pastoralist areas and conditions make it difficult to estimate camel milk production. Camel milk is considered to have anti-cancer, hypo-allergic and anti-diabetic properties. A high content of unsaturated fatty acids contributes to its overall dietary quality. The low quantity of β-casein and the lack of β-lactoglobulin are linked to the hypo-allergic effect of camel milk. Although the Bactrian camel is an important domestic animal in Mongolia, few studies have focused on Bactrian camel milk in this country. Our study characterizes the amino acid and fatty acid compositions of Bactrian camel milk collected from several geographical areas in Mongolia. The findings establish a theoretical basis for additional studies on the composition of milk from Bactrian camels in Mongolia.

**Abstract:**

Camel milk is considered as an essential source of nutrition for desert people. However, few studies have investigated how geography affects Bactrian camel milk in Mongolia. In this study, we evaluated the differences in gross composition, fatty acid composition, and amino acid composition among Bactrian camel milk samples collected from 102 Bactrian camels in five different Mongolian regions. The proportion of long-chain fatty acids, out of total fatty acids, was high in all samples of Bactrian camel milk. The primary fatty acids detected in the samples were palmitic acid (23.99–30.72%), oleic acid (17.21–24.24%), and stearic acid (11.13–16.49%), while the dominant amino acids were leucine, lysine, valine, and aspartic acid. Cysteine was the least common amino acid detected in the Bactrian camel milk samples. Considerable differences in the fatty acid and amino acid compositions were observed among Bactrian camel milk from different regions of Mongolia. The findings suggest that geography strongly affects the composition of camel milk.

## 1. Introduction

Camels are divided into two species: *Camelus dromedaries* (the one-humped dromedary camel) and *Camelus bactrianus* (the two-humped Bactrian camel). Bactrian camels inhabit the semi-arid and rocky mountainous regions, along with the flat regions, of Kazakhstan, Iran, Russia, Mongolia, and China. *C. dromedaries* are domesticated in Australia, the middle east, northern and eastern Africa, and southwest Asia. Of all the Bactrian camel habitat, over 90% is found in Inner Mongolia, China, the Gobi desert in Mongolia, and the desert steppes of Kazakhstan. Camels are known as a good milk source, and camels provide a vital source of milk in hot and arid regions. For people residing in arid and semiarid regions, camel milk provides an excellent source of protein.

Compared to cow milk, camel milk contains higher contents of minerals (calcium, iron, magnesium, copper, zinc, and potassium) and vitamins (VA, B2, E, and C) [1,2]. In addition, the fat in camel milk consists primarily of polyunsaturated fatty acids, and the cholesterol content of camel milk is low [3]. Similar to human milk, camel milk contains α-lactalbumin but lacks β-lactoglobulin and has a low content of β-casein [4,5]. Compared to the milk of dromedary camels, Bactrian camel milk has higher contents of protein, sugar, and fat, along with a lower lactose content [6]. Thus, lactase-deficient children with allergies to cow milk are able to tolerate camel milk [7]. Camel milk can exhibit antibacterial properties, which is similar to the effect of donkey milk [8]. Camel milk also exhibits immunological, anti-carcinogenic, anti-diabetic, anti-cancer, and anti-hypertensive properties [3], along with protective effects against heavy metal toxicity [9] and infections caused by viruses and bacteria [10]. Camel milk has been used to treat diseases such as dropsy, jaundice, tuberculosis, asthma, and leishmaniasis in several countries including India, Russia and Sudan [11]. Camel milk was also reported to protect against non-alcoholic fatty liver disease, insulin-dependent diabetes mellitus, and autism [12,13,14].

Although the Bactrian camel is an important domestic animal in Mongolia, few studies have focused on Bactrian camel milk in this country. Accordingly, it remains necessary to systematically study Bactrian camel milk in Mongolia. This study characterizes the amino acid and fatty acid compositions of Bactrian camel milk collected from several geographical areas in Mongolia. The results provide a basic database of the gross composition and acid composition of Bactrian camel milk from different regions of Mongolia.

## 2. Materials and Methods

Milk samples were collected from 102 Bactrian camels in five different locations in Mongolia (Table 1) during November and December 2016. All camels, with a similar parturition day, 3 to 6 years of age were selected for this study. All samples were obtained by manual milking. After collection, the milk samples were packed in ice and transported to the laboratory, where they were stored at −20 °C. The experiment was conducted according to the animal ethics guidelines of the Key Laboratory of Dairy Biotechnology and Bioengineering, and approved by the Animal Ethics Committee of Inner Mongolia Agricultural University.

### 2.1. Measurement of Gross Milk Composition

The milk samples were dried at 105 °C in a forced draft oven. After drying to a constant weight, total solids were determined using the gravimetric method. The Rȍse–Gottlieb method was used to determine the fat percentage of each milk sample, while the gravimetric method was employed to determine the ash content [15]. The lactose content was calculated by subtracting other solid components from total solids. The protein content was determined using the micro-Kjeldahl method, and total proteins were obtained by multiplying the nitrogen percentage by a factor of 6.38 [16].

### 2.2. Measurement of Amino Acid Content

After transferring to 50-mL cuvettes, sample aliquots containing approximately 800 mg of protein were mixed with 15 mL of 6 M HCl. The cuvettes were sealed, and the samples were hydrolyzed under nitrogen for 22–24 h at 110 °C. After transferring the hydrolysates in 50-mL volumetric flasks, the hydrolysates were mixed with 9 mL of 6 M NaOH followed by dilution with 0.02 M HCl. Subsequently, the samples were filtered and then loaded into an amino acid analyzer (Hitachi L-8900; Tokyo, Japan) for analysis.

### 2.3. Measurement of Fatty Acid Content

For this analysis, 2 mg of milk fat was dissolved in 4 mL N-hexane/isopropanol followed by direct methylation for 15 min using 2 mL NaOCH_3_/methanol. After cooling, successively, in a water bath at 50 °C for 15 min, 2 mL of hydrochloric acid/methanol solution was added. The temperature of the water bath was raised to 80 °C for 1.5 h. Then, 6 mL of a 50% solution of N-hexane in water was added after cooling, and the fatty acid methyl esters were separated by static stratification. The supernatant containing the fatty acids was analyzed by gas chromatography with flame ionization detection. The chromatograph was equipped with an autosampler and an ionic liquid capillary column (PE Clarus 680, 100 m × 0.25 mm × 0.2 μm). The flow rate of nitrogen gas was 1.0 mL/min, and the sample split ratio was 10:1. The column temperature program was as follows: heat to 120 °C at 5 min; hold at 140 °C for 3 min; heat to 240 °C at 1.5 °C/min; hold at 240 °C for 13 min; heat to 245 °C at 20 °C/min; and hold at 245 °C for 6 min.

### 2.4. Statistical Analysis

Data were analyzed using the Kruskal–Wallis rank sum test in the program R. Differences between groups were determined using Welch’s test with correction for false discovery using the Benjamini–Hochberg method. Principal component analysis (PCA) was used to assess correlation between studied parameters.

## 3. Results and Discussion

### 3.1. Gross Composition of Bactrian Camel Milk

The gross compositions of Bactrian camel milk samples collected from the five different regions are shown in Table 2. There is a difference in fat, protein, total solids, and ash content among the considered regions, but no significant difference. Fat content ranged from 4.64–5.7%, protein content ranged from 3.63–3.82%, lactose content ranged from 5.21–5.50%, total solids ranged from 9.52–10.08%, and ash content ranged from 0.73–0.92%. The cross composition of camel milk has been investigated in different parts of the world, including Saudi Arabia, Egypt, China, Kazakhstan, and Mongolia [3,17]. The composition of protein, lactose, fat, ash and total solids of Chinese Bactrian camel milk were reported to range from 3.55–4.45%, 4.23–4.92%, 4.83–5.71%, 0.66–0.94%, and 14.17–15.4%, respectively [17], in agreement with the fat, protein, and ash contents determined for Bactrian camel milk in this study. In this study, the gross composition of milk samples from different locations in Mongolia did not significantly vary in terms of fat, lactose, protein, total solids, and ash content (*p* > 0.05). However, the lactose contents determined herein were higher than the content previously reported for Bactrian camel milk [17]. These variations in the relative proportions of milk components might be attributed to factors such as breed, diet, season, age, stage of lactation, health status, milking frequency, and milking system [18,19].

### 3.2. Amino Acid Composition (%) of Bactrian Camel Milk

Table 3 shows the concentrations of 17 amino acids in the Bactrian camel milk samples from different locations. The amino acid compositions determined in this work are consistent with the findings of previous studies [3,17,20]. Among amino acids in this study, glutamic acid (0.79–0.94%) was the highest in concentration followed by proline. The contents of leucine (0.40–0.47%), lysine (0.29–0.35%), valine (0.25–0.29%), and aspartic acid (0.24–0.29%) in camel milk were also high, whereas cysteine (0.05–0.06%) was the least abundant amino acid. Dispensable amino acids (1.71–2.03%) were more abundant in the milk samples compared to indispensable amino acids (2.13–2.50%). The contents of indispensable, dispensable and total amino acids also differed significantly between geographic regions, with the milk of Tsogtovoo containing the highest fatty acid content (*p* < 0.05). In particular, milk from Tsogtovoo had higher amino acid contents compared to milk from Sharga, including histidine, isoleucine, lysine, phenylalanine, threonine and valine (*p* < 0.05). Among sampling locations, the amino acid contents were the lowest in Hovd and the highest in Tsogtovoo.

While Chinese Bactrian and dromedary camel milk were reported to have similar amino acid compositions to bovine milk, the glycine and cysteine contents in dromedary milk casein were significantly lower than those in bovine milk [17,21]. Rafiq et al. reported high leucine contents in cow, camel, and buffalo milk caseins, with glutamic acid and proline being the predominant non-essential amino acids [20]. In this study, glutamic acid was the most abundant amino acid, similar to the results reported for dromedary camel, cow, sheep, and goat milk [22,23]. The amino acid concentrations determined in this study are in partial agreement with previous reports. The observed differences in the amino acid composition of camel milk between geographic regions may be attributed to differences in diet. The total protein and amino acid concentrations were significantly higher in the milk samples from Tsogtovoo and Bulgan compared to the samples from the other three locations. However, with the exception of cysteine and glutamic acid, the relative contribution of each individual amino acid to the total amino acid content was consistent among regions (Figure 1). Our amino acid analysis revealed large differences in amino acid profiles between geographic regions.

### 3.3. Fatty Acid Composition (%) of Bactrian Camel Milk

Table 4 shows the fatty acid compositions of the Bactrian camel milk fat corresponding to the five different geographic regions. For all locations, the most abundant fatty acids in the samples were C16:0 (23.99–30.72%), C18:0 (11.13–16.49%), C18:1n9c (15.79–22.42%), C14:0 (10.18–13.66%), and C16:1 (5.37–7.53%), in agreement with a past study [24]. Significant differences were observed among locations (*p* < 0.05). In all groups, C16:0 (23.99–30.72%) was the most abundant fatty acid. However, the contents of C16:0 and C14:0 in Hovd and Tseel milk samples were significantly different to those of samples from other locations (*p* < 0.05). Compared to other groups, the content of C18:1n9c in the Hovd group was significantly different (*p* < 0.05). The contents of C18:0 in the Hovd and Sharga groups were significantly different than those of the other groups (*p* < 0.05). The Bulgan and Tsogtovoo samples had higher concentrations of C16:1 compared to samples from the other three locations (*p* < 0.05). The total contents of C4:0–C8:0 fatty acids were low in all groups, in agreement with a previous study on camel milk [17]. Saturated fatty acids (57.53–65.24%) were more abundant in this study than unsaturated fatty acids, consistent with the observations of Konuspayeva et al. [24]. Chilliard et al. considered the effects of three factors (region, season, and species) on the fatty acid composition of milk, despite the role of feed type and physiological stage [25]. The differences in fatty acid composition in this study can also be explained by the different geographic regions.

The proportion of saturated fatty acids (SFAs) was lower in the samples from Hovd and Bulgan compared to the samples from the other locations (Table 5). C16:0, C18:0, and C14:0 were the predominant SFAs, similar to those reported in Bactrian and dromedary camel milk from China [17]. Significant differences between groups were observed in both SFA and unsaturated fatty acid (UFA) contents. The contents of monounsaturated fatty acids (MUFAs) and polyunsaturated fatty acids (PUFAs) were significantly higher in the Hovd group compared to other groups. The most abundant MUFA in this study was C18:1, which has also been reported in milk from Bactrian camels [24], dromedary camels [3], and cows [26]. The milk samples from all locations in this study were enriched in n-3 fatty acids, and 18:2n-6 was the most abundant n-6 fatty acid. The ratios of n-6 to n-3 fatty acids determined for all groups were close to the ideal range suggested for the human diet (2:1–4:1) [27]. Among all groups, the Sharga group had the lowest n-6/n-3 ratio (Table 5).

Among fatty acids in milk, 60% originate from the uptake of fatty acids from circulating blood by the mammary glands, while the other 40% result from de novo synthesis within the mammary glands [25]. Almost all of the C4:0–C14:0 odd-chain fatty acids and approximately half of the C16:0 in milk fat originate from de novo synthesis in the mammary glands [28]. The remaining C16:0 and all of the long-chain fatty acids (LCFAs) originate from lipids in circulating blood, with n-6 and n-3 fatty acids being taken up from plasma [29]. The specific breed of cow or goat has been reported to affect the contents of C14:1c9, C16:1c9, and C18:1c9 in the mammary glands [30,31]. In this study, The results showed that the geographic region may also affect the synthesis of fatty acids in the mammary glands [32].

The amino acid contents in the 112 Bactrian camel milk samples were analyzed by PCA. Figure 2 shows the distribution of traits in each of the first two components, PC1 and PC2, which explained 93.3% and 3.1% of the variability, respectively. The lines in Figure 2 indicate eigenvectors that represent the strength and direction of each principal component [33]. Together, PC1 and PC2 explained 96.4% of all variability. While PC1 had a positive association with all variables, PC2 was mainly influenced by the proportions of cysteine and tyrosine, which were positively associated with PC2. As shown in Figure 2, the Bactrian camel milk from Bulgan was clearly separated from the samples from Hovd and Tseel in the PCA results.

The PCA results for the fatty acid data are presented in Figure 3. Collectively, the first two retained components (PC1 and PC2) represented 52% of total variance, with 30.2% of total variance represented by PC1. PUFA, C20:0, C18:3n-3, n-6FA, n-3FA, C22:0, C18:2n6, C21:0, and C18:0 were the dominant fatty acid groups in PC1 and were positively correlated with PC1. In contrast, C16:1, C15:0, C8:0, C14:1, C20:3n-3, SCFA, and MCFA were negatively correlated with PC1. PC2 represented 11.8% of total variance and was highly influenced by MUFA, C17:1, OCFA, C18:1n-9c, and C20:3n-3. SFA, and C112:0, and C18:0 were negatively correlated with PC2. As shown in Figure 3, the Bactrian camel milk from Bulgan was clearly separated from the milk samples from Hovd, Sharga, and Tseel in terms of fatty acid composition. Numerous controlled and farm experiments have been conducted to evaluate the effects of farming systems on the fatty acid compositions of ruminant milk [30,34,35]. In this study, only milk from Bulgan was clearly separated from the other groups in the PCA results for fatty acids. Bactrian camels inhabit areas with arid, cold climates (long, cold winters and short summers) in Mongolia. These areas have scarce water sources and sparse vegetation, and the diets of camels in these regions are relatively simple [36]. At present, Mongolian camels have not been identified, apart from one breed. Camels are not artificially supplemented and always naturally graze. The camel’s diet has certain differences in different areas, which may be the main reason for the different nutrients in its milk. This implies that the environment of the host has stronger effects on camel milk composition. The Umnugovi aimag Bulgan sum and Umnugovi aimag Tsogtovoo sum Bor teeg bag Erdenet olgoi environments are more suitable for camel growth.

Figure 4 presents the PCA results obtained using both the amino acid and fatty acid data. The variables were reduced into two primary principal components: PC1, which explained 36.2% of total variability; and PC2, which explained 14.2% of total variability. PC1 was highly influenced by and was positively associated with all amino acid variables, while PC1 was negatively correlated with C16:1, C15:0, C8:0, C14:1, C20:3n-3, SCFA, and MCFA. PC2 was mainly influenced by C18:2n6c, C18:0, C18:3n3, C20:0, C21:0, and C22:0, whereas PC2 was negatively correlated with C16:1, C8:0, C6:0, and C12:0. The milk samples clustered into two groups (Figure 4): (i) Bulgan and Tsogtovoo samples; and (ii) Hovd, Sharga, and Tseel samples. The PCA scatter plot shown in Figure 4 can effectively distinguish between the Bactrian camel milk samples from different regions. Therefore, the combination of fatty acid and amino acid compositions could be used to differentiate camel milk from the five different regions in Mongolia in this study.

## 4. Conclusions

Camel milk, which is a major type of milk consumed by people residing in arid regions, provides nutrition in the form of proteins, amino acids, and fatty acids. This study evaluated the gross compositions, amino acid compositions, and fatty acid compositions to differentiate between Bactrian camel milk collected from five different regions of Mongolia. The data suggest a strong effect of geography on the composition of camel milk. The results indicate that the PCA of both fatty acid and amino acid compositions can be used to differentiate Bactrian camel milk from different regions in Mongolia. The Bactrian camel milk samples were characterized by large proportions of LCFAs, and the main amino acids in the milk samples were leucine, lysine, valine, and aspartic acid. The findings establish a theoretical basis for additional studies on the composition of milk from Bactrian camels in Mongolia.

## Figures and Tables

**Figure 1 animals-09-00890-f001:**
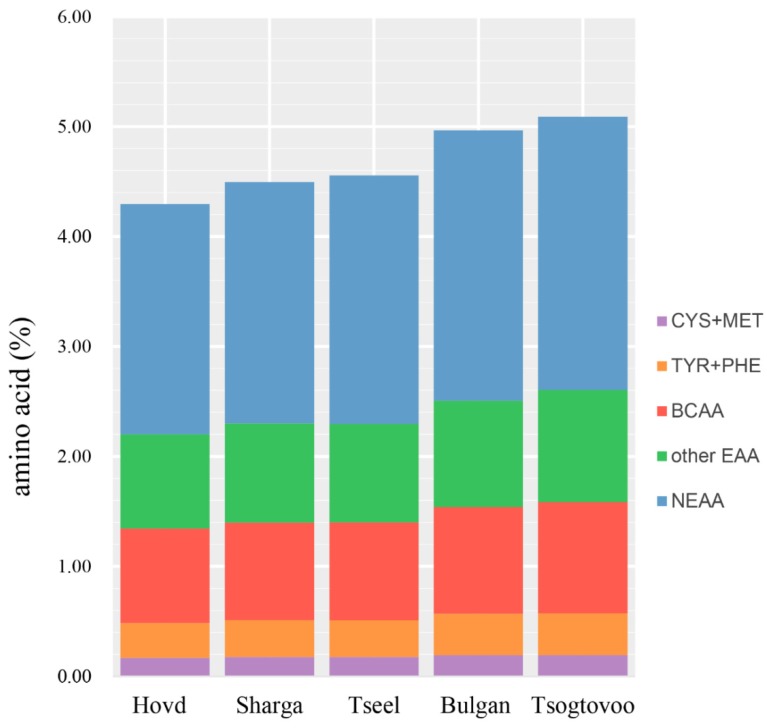
Relative contributions of individual amino acids to total amino acids for camel milk sampled from different regions in Mongolia. CYS + MET, cystine + methionine; TYR + PHE, tyrosine + phenylalanine; BCAA, branched-chain amino acids; Other EAA, other essential amino acid; NEAA, non-essential amino acid.

**Figure 2 animals-09-00890-f002:**
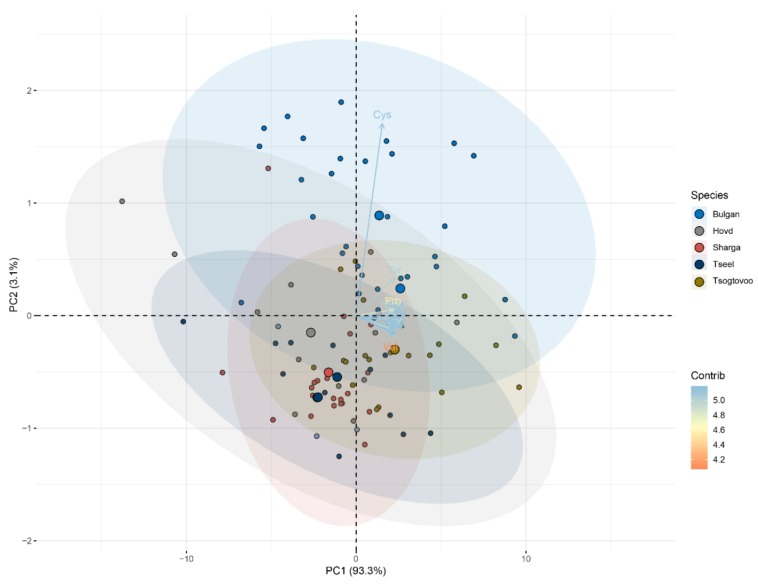
Results of PCA analysis of the amino acid compositions of camel milk from five regions of Mongolia.

**Figure 3 animals-09-00890-f003:**
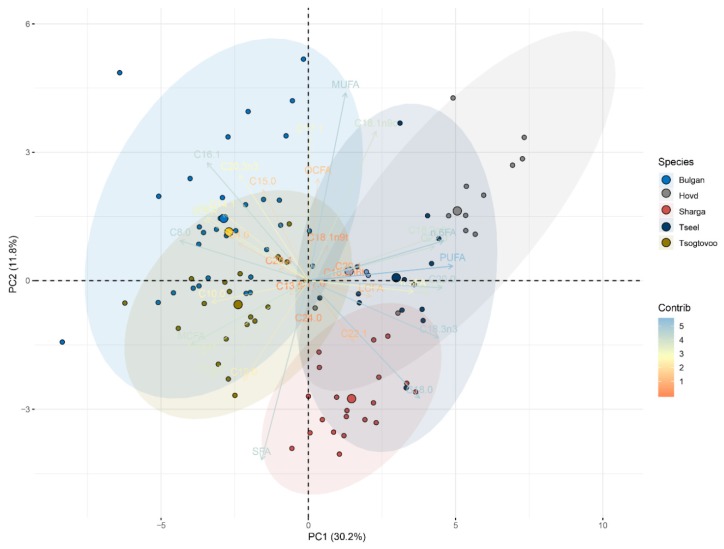
Results of PCA analysis of the fatty acid compositions of camel milk from five regions of Mongolia.

**Figure 4 animals-09-00890-f004:**
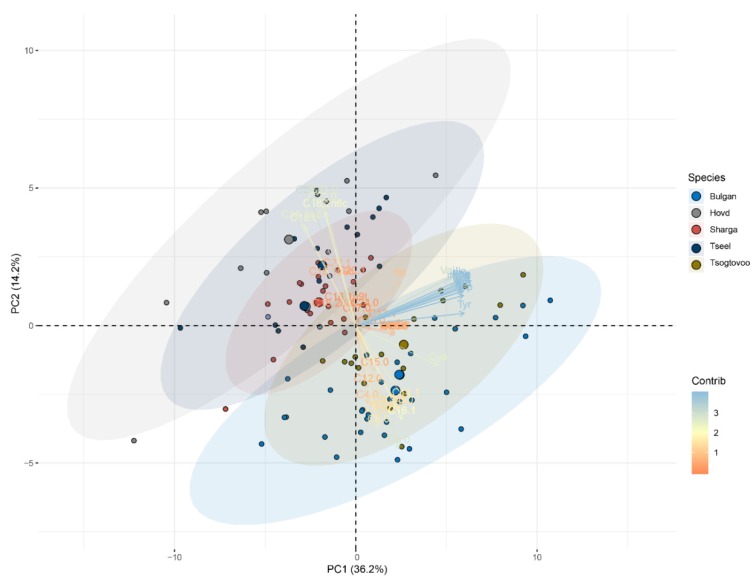
Results of PCA analysis of the fatty acid and amino acid compositions of camel milk from five regions of Mongolia.

**Table 1 animals-09-00890-t001:** Geographic locations of Bactrian camel milk sampling in Mongolia.

Location	Number of Samples
Hovd aimag Zereg sum (Hovd)	13
Gobialtai aimag Sharga sum Sonduult (Sharga)	21
Gobialtai aimag Tseel sum Buuriin gol (Tseel)	16
Umnugovi aimag Bulgan sum (Bulgan)	32
Umnugovi aimag Tsogtovoo sum Bor teeg bag Erdenet olgoi (Tsogtovoo)	20

**Table 2 animals-09-00890-t002:** Gross compositions of Bactrian camel milk sampled from different locations in Mongolia.

Region	Fat (%)	Protein (%)	Lactose (%)	Total Solids (%)	Ash (%)
Hovd	6.24 ± 1.29	3.65 ± 0.40	5.24 ± 0.59	9.62 ± 1.08	0.73 ± 0.09
Sharga	5.14 ± 0.61	3.63 ± 0.49	5.24 ± 0.73	9.60 ± 1.32	0.89 ± 0.07
Tseel	5.01 ± 0.85	3.62 ± 0.49	5.21 ± 0.71	9.52 ± 1.90	0.89 ± 0.09
Bulgan	5.40 ± 1.30	3.74 ± 0.42	5.40 ± 0.60	9.89 ± 1.11	0.92 ± 0.09
Tsogtovoo	6.01 ± 1.41	3.82 ± 0.17	5.50 ± 0.23	10.08 ± 0.45	0.84 ± 0.07

Each value is presented as the mean ± standard deviation.

**Table 3 animals-09-00890-t003:** Amino acid compositions of Bactrian camel milk from different locations in Mongolia.

Amino Acid	Hovd	Sharga	Tseel	Bulgan	Tsogtovoo
Histidine	0.10 ± 0.02 ^b^	0.11 ± 0.01 ^b^	0.11 ± 0.01 ^b^	0.09 ± 0.02 ^a,b^	0.12 ± 0.01 ^a^
Isoleucine	0.21 ± 0.05 ^a,b^	0.22 ± 0.02 ^a^	0.22 ± 0.03 ^a,b^	0.24 ± 0.04 ^a,b^	0.25 ± 0.03 ^b^
Leucine	0.40 ± 0.09 ^a,b^	0.42 ± 0.04 ^a^	0.41 ± 0.06 ^a,b^	0.45 ± 0.08 ^a,b^	0.47 ± 0.06 ^b^
Lysine	0.29 ± 0.05 ^b^	0.31 ± 0.02 ^b^	0.30 ± 0.04 ^b^	0.33 ± 0.05 ^a,b^	0.35 ± 0.03 ^a^
Methionine	0.12 ± 0.02 ^a^	0.13 ± 0.01 ^a,b^	0.13 ± 0.02 ^a,b^	0.13 ± 0.02 ^a,b^	0.14 ± 0.02 ^b^
Phenylalanine	0.17 ± 0.03 ^a,b^	0.18 ± 0.01 ^b^	0.18 ± 0.02 ^b^	0.19 ± 0.03 ^a,b^	0.20 ± 0.02 ^a^
Threonine	0.18 ± 0.03 ^b^	0.18 ± 0.01 ^b^	0.19 ± 0.02 ^a,b^	0.20 ± 0.03 ^a,b^	0.21 ± 0.02 ^a^
Valine	0.25 ± 0.05 ^a,b^	0.25 ± 0.06 ^a^	0.26 ± 0.03 ^a,b^	0.27 ± 0.04 ^a,b^	0.29 ± 0.03 ^b^
Total EAA	1.71 ± 0.35 ^b^	1.79 ± 0.17 ^b^	1.78 ± 0.24 ^a,b^	1.94 ± 0.32 ^a,b^	2.03 ± 0.22 ^a^
Alanine	0.09 ± 0.02 ^b^	0.09 ± 0.01 ^b^	0.09 ± 0.01 ^a,b^	0.10 ± 0.02 ^a,b^	0.11 ± 0.01 ^a^
Arginine	0.15 ± 0.03 ^b^	0.16 ± 0.01 ^b^	0.16 ± 0.02 ^b^	0.17 ± 0.03 ^a,b^	0.18 ± 0.02 ^a^
Aspartic acid	0.24 ± 0.04 ^a,b^	0.25 ± 0.02 ^a^	0.26 ± 0.03 ^a,b^	0.28 ± 0.05 ^b,c^	0.29 ± 0.03 ^c^
Glutamic acid	0.79 ± 0.16 ^b^	0.83 ± 0.06 ^b^	0.84 ± 0.11 ^a,b^	0.91 ± 0.15 ^a,b^	0.94 ± 0.09 ^a^
Glycine	0.05 ± 0.01 ^b^	0.05 ± 0.01 ^b^	0.05 ± 0.01 ^b^	0.05 ± 0.01 ^a,b^	0.06 ± 0.01 ^a^
Serine	0.16 ± 0.03 ^b^	0.18 ± 0.01 ^b^	0.19 ± 0.02 ^a,b^	0.20 ± 0.03 ^a,b^	0.21 ± 0.02 ^a^
Tyrosine	0.15 ± 0.03 ^a^	0.16 ± 0.01 ^a^	0.16 ± 0.02 ^a^	0.18 ± 0.03 ^b^	0.18 ± 0.02 ^b^
Proline	0.40 ± 0.09 ^b^	0.43 ± 0.03 ^b^	0.48 ± 0.07 ^a,b^	0.49 ± 0.08 ^b^	0.46 ± 0.06 ^a,b^
Cystine	0.05 ± 0.01 ^b,c^	0.05 ± 0.01 ^c^	0.05 ± 0.01 ^c^	0.06 ± 0.01 ^a^	0.05 ± 0.01 ^a,b^
Total NEAA	2.13 ± 0.43 ^a,b^	2.19 ± 0.17 ^a^	2.26 ± 0.30 ^a,b^	2.50 ± 0.42 ^b^	2.49 ± 0.42 ^b^
TAA	3.85 ± 0.77 ^a,b^	3.98 ± 0.33 ^a^	4.05 ± 0.54 ^a,b^	4.44 ± 0.74 ^b^	4.52 ± 0.47 ^b^

TAA, total amino acid; EAA, essential amino acid; NEAA, non-essential amino acid. Values are given as mean ± SEM. ^a,b,c^ Data with different superscript letters in the same column of variety, respectively, indicate significant difference.

**Table 4 animals-09-00890-t004:** Fatty acid compositions of Bactrian camel milk from different locations in Mongolia.

Fatty Acid	Hovd	Sharga	Tseel	Bulgan	Tsogtovoo
C4:0	0.02 ± 0.04 ^a^	0.01 ± 0.01 ^a^	0.01 ± 0.02 ^a^	0.04 ± 0.03 ^b^	0.04 ± 0.05 ^b^
C6:0	0.20 ± 0.11 ^b,c^	0.10 ± 0.02 ^a^	0.12 ± 0.06 ^a,b^	0.30 ± 0.16 ^c^	0.22 ± 0.11 ^b,c^
C8:0	0.09 ± 0.08 ^a^	0.13 ± 0.03 ^a^	0.10 ± 0.06 ^a^	0.36 ± 0.16 ^b^	0.29 ± 0.16 ^b^
C10:0	0.01 ± 0.03 ^a^	0.11 ± 0.06 ^a,^^b^	0.05 ± 0.06 ^a^	0.16 ± 0.09 ^b,c^	0.18 ± 0.07 ^c^
C12:0	0.81 ± 0.09 ^a^	1.09 ± 0.14 ^a^	0.83 ± 0.08 ^b^	0.98 ± 0.22 ^b^	1.00 ± 0.08 ^b^
C13:0	0.01 ± 0.02	0.03 ± 0.04	-	0.03 ± 0.03	0.06 ± 0.20
C14:0	10.18 ± 1.30 ^a^	12.64 ± 1.57 ^b^	12.24 ± 0.91 ^a^	13.15 ± 2.46 ^b^	13.66 ± 0.98 ^b^
C15:0	1.94 ± 0.17 ^a^	1.71 ± 0.24 ^a^	2.14 ± 0.58 ^b^	2.23 ± 0.27 ^b^	1.94 ± 0.14 ^a^
C16:0	23.99 ± 1.47 ^a^	30.08 ± 1.82 ^b^	27.40 ± 6.82 ^a^	30.48 ± 2.11 ^b^	30.72 ± 1.72 ^b,c^
C17:0	1.16 ± 0.09 ^c^	1.21 ± 0.10 ^b,c^	1.46 ± 0.12 ^a^	1.29 ± 0.17 ^b^	1.22 ± 0.12 ^b,c^
C18:0	16.18 ± 1.91 ^a^	16.49 ± 1.18 ^a^	14.99 ± 2.01 ^a^	11.13 ± 2.46 ^b^	12.27 ± 1.59 ^b^
C20:0	1.72 ± 0.37 ^c^	0.89 ± 0.28 ^b,c^	1.09 ± 0.25 ^c^	0.39 ± 0.12 ^a^	0.56 ± 0.06 ^a,b^
C21:0	0.56 ± 0.21 ^c^	0.37 ± 0.05 ^b^	0.46 ± 0.06 ^c^	0.13 ± 0.09 ^a^	0.39 ± 0.12 ^b,c^
C22:0	0.64 ± 0.31 ^a^	0.34 ± 0.11 ^a,b^	0.55 ± 0.17 ^a^	0.22 ± 0.11 ^b,c^	0.15 ± 0.09 ^c^
C24:0	0.02 ± 0.05	0.05 ± 0.07	0.06 ± 0.09	0.03 ± 0.05	0.06 ± 0.07
C14:1	0.27 ± 0.17 ^a^	0.42 ± 0.10 ^a^	0.42 ± 0.09 ^a^	0.62 ± 0.20 ^b^	0.58 ± 0.14 ^b^
C16:1	5.86 ± 0.68 ^a^	5.37 ± 0.60 ^a^	5.72 ± 1.74 ^a^	7.53 ± 1.74 ^b^	7.40 ± 0.72 ^b^
C17:1	0.66 ± 0.23 ^b,c^	0.56 ± 0.04 ^a^	0.76 ± 0.07 ^b,c^	0.79 ± 0.16 ^c^	0.63 ± 0.23 ^b^
C21:1	0.10 ± 0.12 ^b,c^	0.13 ± 0.08 ^a^	0.15 ± 0.10 ^b,c^	0.11 ± 0.07 ^c^	0.13 ± 0.08 ^b^
C22:1	0.29 ± 0.43 ^b,c^	0.30 ± 0.15 ^a^	0.36 ± 0.16 ^b,c^	-	0.48 ± 0.25 ^b^
C18:1n-9t	1.82 ± 0.36 ^c^	1.42 ± 0.20 ^b,c^	1.39 ± 0.28 ^b,c^	1.08 ± 1.93 ^a^	1.20 ± 0.29 ^b^
C18:1n-9c	22.42 ± 2.02 ^a^	15.79 ± 1.60 ^b^	17.87 ± 1.74 ^c^	18.13 ± 2.52 ^b^	17.06 ± 1.88 ^c^
C18:2n-6t	0.19 ± 0.18	0.24 ± 0.40	0.27 ± 0.28	0.15 ± 0.11	0.14 ± 0.09
C18:2n-6c	2.54 ± 0.27 ^c,d^	1.92 ± 0.38 ^b,c^	2.21 ± 0.21 ^c,d^	1.73 ± 0.23 ^a,b^	1.68 ± 0.19 ^a^
C18:3n-3	1.13 ± 0.15 ^a^	1.21 ± 0.19 ^a^	1.03 ± 0.12 ^a^	0.61 ± 0.17 ^b^	0.46 ± 0.06 ^b^
C20:4	-	0.03 ± 0.04	0.01 ± 0.02	0.06 ± 0.08	0.03 ± 0.05

Each value is presented as the mean ± standard deviation. ^a,b,c,d^ Data with different superscript letters in the same column of variety, respectively, indicate significant difference.

**Table 5 animals-09-00890-t005:** Fatty acid profiles of Bactrian camel milk samples from different locations in Mongolia.

Fatty Acid	Hovd	Sharga	Tseel	Bulgan	Tsogtovoo
SFA	57.53 ± 3.37 ^a^	65.24 ± 2.67 ^b^	61.12 ± 7.19 ^b^	60.90 ± 4.23 ^a^	62.75 ± 2.41 ^b^
MUFA	34.15 ± 2.77 ^a^	26.14 ± 2.13 ^b^	29.16 ± 1.79 ^c^	30.13 ± 3.47 ^c^	29.29 ± 2.08 ^c^
PUFA	4.42 ± 0.61 ^a^	3.76 ± 0.44 ^a^	3.97 ± 0.29 ^a^	2.94 ± 0.51 ^b^	2.70 ± 0.36 ^b^
SCFA	0.20 ± 0.15 ^a,b^	0.11 ± 0.02 ^a^	0.13 ± 0.07 ^a^	0.34 ± 0.17 ^c^	0.26 ± 0.13 ^b,c^
MCFA	11.35 ± 1.38 ^a^	14.41 ± 1.73 ^b,c^	13.66 ± 1.09 ^a,b^	15.29 ± 2.67 ^c^	15.77 ± 1.29 ^c^
LCFA	81.26 ± 2.15 ^a^	78.10 ± 1.14 ^a,b^	77.52 ± 7.80 ^a^	76.34 ± 1.29 ^c^	76.51 ± 1.51 ^b,c^
OCFA	4.32 ± 0.48 ^a,b^	3.87 ± 0.34 ^a^	4.82 ± 0.60 ^c^	4.47 ± 0.40 ^b,c^	4.23 ± 0.42 ^a,b^
n-6 FA	2.73 ± 0.35 ^a^	2.16 ± 0.30 ^b^	2.48 ± 0.21 ^a,b^	1.87 ± 0.23 ^c^	1.82 ± 0.23 ^c^
n-3 FA	1.14 ± 0.15 ^a^	1.21 ± 0.19 ^a^	1.03 ± 0.12 ^a,b^	0.88 ± 0.27 ^b^	0.46 ± 0.06 ^c^
n-6/n-3	2.42 ± 0.29 ^c^	1.80 ± 0.24 ^b^	2.44 ± 0.25 ^c^	2.34 ± 1.92 ^b,c^	3.97 ± 0.34 ^a^

^a,b,c^ Data with different superscript letters in the same column of variety, respectively, indicate significant difference.

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
