# Peer review of "Effects of Geographic Region on the Composition of Bactrian Camel Milk in Mongolia"

_animals, 2019, doi:10.3390/ani9110890_

Round 1
Reviewer 1 Report
This manuscript shows several criticisms.
the first is the absence of information about the animals and other points. Regarding the animals: what is the age, the race, the stage of lactation...What the animals fed? What are the environmental conditions? What is the way for milk collection?
these are all points that can affect milk composition...so we cannot consider the site of collection as a factor affecting the milk composition.
In addition, the tables are poor presented. It needs a average value for each parameter, a general source of variation as rmse or sem and the exact p value.
Author Response
We must thank you for the critical feedback. We feel lucky that our manuscript went to these reviewers as the valuable comments from them not only helped us with the improvement of our manuscript. Based on these comments and suggestions, we have made careful modifications on the original manuscript. we are now sending the revised article for your re-consideration . Please see our point to point responses to all your comments below, and the corresponding revisions in the body of manuscript, both marked in red.
We hope that these revisions are satisfactory and that the revised version will be acceptable for publication.

Reviewer 2 Report
“Effects of geographic region on the composition of Bactrian camel milk in Mongolia ” by He Jing et al., presents an evaluation of the differences in gross composition, fatty acid composition, and amino acid composition among Bactrian camel milk samples in Mongolia. A total of 102 Bactrian camels were used in this study, making this study quite comprehensive. Few studies have focused on Bactrian camel milk in Mongolia. The article presentation is good. However, some issues still need to be improved.
Tables require further information. For example, the meanings of superscript letters in Table 1 are not provided. Fatty acid correlation analysis suggested deletion. The paper does not address whether the differences in milk composition are thought to be related to the presence of different camel breeds or differences in other factors between regions. Consider adding additional discussion related to the specific factors responsible for the geographic differences in camel milk. It is suggested that the authors check the English writing.Author Response
We must thank you and all other reviewers for the critical feedback. We feel lucky that our manuscript went to these reviewers as the valuable comments from them not only helped us with the improvement of our manuscript. Based on these comments and suggestions, we have made careful modifications on the original manuscript. we are now sending the revised article for your re-consideration. Please see our point to point responses to all your comments below, and the corresponding revisions in the body of manuscript, both marked in red.
We hope that these revisions are satisfactory and that the revised version will be acceptable for publication.

Reviewer 3 Report
The manuscript deals with a very interesting investigation on the composition of Bactrian camel milk in five regions of Mongolia, although there is some room for further improvements.
About it:
a) a brief description of the characteristics of the pastures in the study areas could have made the work even more interesting, since in herbivores diet influence fatty acid profile in milk. b) were ,or not, animals fed on supplementation? c) do the Bactrians emploied in the study belong to the same breed?I recommend re-checking carefully the manuscript and revising English.
------------------------------------------------------------
line 22 - write (23.99–30.72%), oleic acid (17.21– 24.24%), and stearic acid (11.13– 16.49%), (use this notation also for comparisons in Results)
line 39 - write vitamins (A, B2 , E, and C)
line 45 - write Camel milk, similarly to donkey milk [* ** you could refer to bibliography below] also exhibits antibacterial,….
line 58 - write of Mongolia
line 75 - write. Cuvettes were sealed, and samples
line 81 - write. For this analysis 2 mg of milk fat were dissolved in 4 mLN-hexane/isopropanol. Followed direct methylation……
line 83 - write Successively, in water bath at 50℃ for 15 min, 2 mL of hydrochloric acid/methanol solution was added
line 84 - write . Followed the addition of 2 mL hydrochloric acid/methanol…….
line 88 - write a ionic
line 101- write Fat, protein, total solids, and ash contents differ significantly among the considered regions. Moreover, if “contents differ significantly” where are comparisons in table 2?
109-111 -“ In this study, the gross composition of milk samples from different locations in Mongolia did not who (?) significant variations in terms of fat, lactose, protein, ……and ash contents (P > 0.05)”. In contrast with line 101. Please check carefully sub chapter 3.1.
Table 2 report (%) in title and remove from table (see table 3)
Line 150 - “This study provides considerable evidence that the protein content and amino acid composition of human milk are relatively difference across geographic regions.” Not clear, reformulate
Line 239 delete “ruminant”. Citation [34] concerns donkey milk
---------------------------------------------------------------
* Cosentino, C,; Paolino, R.; Musto M.; Freschi, P. Innovative use of jenny milk from sustainable rearing. In Sustainability of Agro-Food and Natural Resource Systems in the Mediterranean Basin; Editor Vastola A; Springer International Publishing Switzerland, 2015, pp. 113-132.
**Labella, C.; Lelario, F.; Bufo, S. A.; Musto, M.; Freschi, P.; Cosentino, C. Optimization and validation of a chromatographic method for quantification of lysozyme in jenny milk. Journal of Food and Nutrition Research 2016, 55, 263-269.
Author Response
We must thank you and all other reviewers for the critical feedback. We feel lucky that our manuscript went to these reviewers as the valuable comments from them not only helped us with the improvement of our manuscript. Based on these comments and suggestions, we have made careful modifications on the original manuscript. we are now sending the revised article for your re-consideration. Please see our point to point responses to all your comments below, and the corresponding revisions in the body of manuscript, both marked in red.
We hope that these revisions are satisfactory and that the revised version will be acceptable for publication.

Round 2
Reviewer 1 Report
dear authors,
thank you for changing your manuscript.